# The Effect of Different Pulse Widths on Lattice Temperature Variation of Silicon under the Action of a Picosecond Laser

**DOI:** 10.3390/mi13071119

**Published:** 2022-07-15

**Authors:** Jianjun Yang, Decheng Zhang, Jinye Wei, Lingling Shui, Xinjin Pan, Guangren Lin, Tiande Sun, Yicheng Tang

**Affiliations:** 1Zhongshan Institute, College of Electron and Information, University of Electronic Science and Technology of China, Zhongshan 528402, China; 15362135441@163.com (D.Z.); 17806270127@163.com (J.W.); xinjpan@163.com (X.P.); 2South China Academy of Advanced Optoelectronics, South China Normal University, Guangzhou 510006, China; shuill@m.scnu.edu.cn; 3Guangdong Huakuai Photonics Technology Co., Ltd., Zhongshan 528400, China; 15766330062@163.com (G.L.); 15913333059@163.com (T.S.); tangyic27@126.com (Y.T.)

**Keywords:** pulse width, picosecond laser, silicon, lattice temperature

## Abstract

In laser processing, due to the short interaction time between an ultrashort pulse laser and silicon, it has been difficult to study the lattice temperature change characteristics of silicon. In this paper, the interaction between a picosecond laser and silicon was studied. Based on the Fokker–Planck equation and two-temperature model (TTM) equation, a simulation model of silicon heating by different pulse-width picosecond lasers was established. The results show that within the range of 15 to 5 ps, the maximum lattice temperature tended to increase first and then decrease with the decreasing pulse width. The watershed was around 7.5 ps. The model error was less than 3.2% when the pulse width was 15 ps and the single pulse energy was 25 μJ.

## 1. Introduction

Silicon is the most basic material for the production of electronic components and optoelectronic devices [1,2,3]. The processing technologies of silicon include diamond grinding, lithography, chemical etching, laser processing, etc. However, diamond grinding can lead to defects such as fragmentation and excessive processing area [4], and the processes of lithography and chemical etching are complex and have poor efficiency [5]. In recent years, picosecond and femtosecond lasers have begun to be used in the silicon industry because of the characteristics of ultra-short pulse width and ultra-high peak power, which can realize the direct writing of integrated optical devices and surface micro-nano structure processing [6,7,8]. Compared with the femtosecond laser, the picosecond laser has a lower cost and more mature manufacturing technology.

The interaction mechanism between the picosecond laser and silicon is complex; the influencing factors include laser wavelength, pulse width, pulse energy and material properties. Due to the difficulty of accessing equipment and conducting experiments, the published experimental data are very limited and low quality; therefore, a theoretical analysis is more beneficial to explore the mechanism of laser-processing materials. When the time scale is a nanosecond or even longer, the classical Fourier heat transfer can describe the heat transfer process inside the material very well [9]. When the picosecond and femtosecond laser interact with materials, the classical Fourier heat transfer does not describe the heat transfer process [10]. The atomic response at the atomic scale can be calculated by means of molecular dynamics. When the picosecond laser irradiates on silicon, a strong nonlinear absorption effect occurs. The ultra-intense photon energy density causes electrons in the valence band to excite to the conduction band to form free electrons, which absorb photon energy and are rapidly heated in a few femtoseconds [11]. The electrons and lattice collide with each other to transfer energy to the lattice, leading to an increase in the lattice temperature. When the lattice temperature reaches the boiling point of the material, it leads to the vaporization removal of the material [12]. The Fokker–Planck equation and TTM can describe the physical process of picosecond and femtosecond laser interaction with materials very well [13,14]. Wang et al. [15] established a finite element model for the femtosecond laser ablation of sapphire using the Fokker–Planck equation and the Drude equation, in order to study the spatio-temporal evolution of free electron concentration. Cai et al. [16] established a multi-photon absorption model of sapphire substrate under the action of a femtosecond laser with different wavelengths. Zhu et al. [17] carried out a numerical study of the picosecond laser micro-grooving of single crystal germanium and compared the simulating ablation depth of different material removal mechanisms. Gan et al. [18] established a hybrid model of the interaction between an ultra-fast pulse laser and semiconductor at the atomic level. Goodarzi et al. [19] carried out a theoretical study of linear chirp effect on femtosecond laser heating silicon using a two-temperature model. Thorstensen et al. [20] established a simulation model of the interaction between a femtosecond laser and silicon and analyzed the effects of different initial substrate temperatures on the ablation threshold. In the application of picosecond laser processing silicon, the pulse width of laser cannot usually be changed, but it affects the lattice temperature of silicon and thus the processing efficiency. Currently, few people have carried out research on the lattice temperature variation of silicon by pulse width.

In this paper, the Fokker–Planck equation was used to simulate the process of free electrons excitation under the action of a picosecond laser, and TTM was used to simulate the process of electron–lattice energy coupling. The two-dimensional simulation model of silicon heating by a picosecond laser with a 1064 nm wavelength was established by COMSOL finite element simulation software, and the effects of different pulse widths on free electron concentration, electron temperature, lattice temperature, absorption coefficient and reflection coefficient were analyzed. The simulating ablation depth is in good agreement with the experiment.

## 2. Theoretical Model and Parameters

The process of ultra-fast pulse laser heating silicon is a non-equilibrium thermal process. Inside materials, the mass of protons is much heavier than that of electrons [21]. When an ultra-fast pulse laser irradiates silicon, the response of ions to the laser-oscillating electric field is weak, and the photon energy is firstly absorbed by the electrons. The electrons in the valence band absorb the photon energy and excite to the conduction band to form free electrons. The free electrons are rapidly heated under the action of the laser-oscillating electric fields, and they transfer energy to the lattice by colliding with it [22]. Therefore, the interior of the material can be regarded as two subsystems of electron and lattice, a method that was first proposed by Kaganov and called the two-temperature model [23]. Later, the two-temperature model was improved and applied to the study of the interaction between an ultrafast pulse laser and semiconductor materials [24].

### 2.1. Free Electrons Excitation

The picosecond laser has an ultrahigh power density. When the laser pulse irradiating on silicon, silicon excites the electrons in the valence band to the conduction band through single-photon absorption and two-photon absorption to generate free electrons, while reducing the free electron concentration through the diffusion effect and auger recombination. The Fokker–Planck equation can describe this physical process, as shown in Equation (1):(1)$$\frac{\partial N}{\partial t}=\nabla \left(D_{0}\nabla N\right)-\gamma N^{3}+\frac{\alpha I}{hv}+\frac{\beta I^{2}}{2hv}$$
where N is the free electron concentration, t is time, $D_{0}$ is the electron diffusivity coefficient, γ is the auger recombination coefficient, α is the single-photon absorption coefficient, β is the two-photon absorption coefficient, h is Plank’s constant and v is the photon frequency [20]. I is the laser power density, which is Gaussian, distributed in both time and space and shown as follows:(2)$$I=\sqrt{\frac{4ln2}{\pi }}\frac{2J}{\left(\pi r_{0}{}^{2}\right)t_{p}}\left(1-R\right)exp(-4ln2\frac{{\left(t-t_{0}\right)}^{2}}{t_{p}{}^{2}})exp(-2\frac{r^{2}}{r_{0}{}^{2}})exp(-\int_{0}^{z}\left(\alpha +\beta I\right)dz)$$
where R is the reflection coefficient, $t_{0}$ is the moment of peak laser power density, $t_{p}$ is the pulse width, J is single pulse energy, z and r are two-dimensional plane coordinates, and $r_{0}$ is the surface spot radius [25,26].

### 2.2. Energy Absorption and Heat Transfer in Electrons

Under the irradiation of a picosecond laser, the free electrons that are excited into the conduction band are rapidly heated by the oscillating electric field, and the heated electrons reach a thermal equilibrium state through electron–lattice energy coupling. This process can be described as:(3)$$\frac{\partial \left(NE_{g}+C_{e}T_{e}\right)}{\partial t}=\nabla \left(K_{e}\nabla T_{e}\right)+\alpha I+\beta I^{2}-\frac{3NK_{b}}{\tau_{e}}\left(T_{e}-T_{l}\right)$$
where $E_{g}$ is the band gap width, $C_{e}$ is the electron heat capacity, $T_{e}$ is the electron temperature, $K_{e}$ is the electron heat transfer coefficient, $K_{b}$ is the Boltzmann coefficient, and $\tau_{e}$ is the electron–lattice relaxation time [27].

### 2.3. Electron–Lattice Energy Coupling

The heated electrons collide with the lattice and transfer energy to it, resulting in a rise in lattice temperature, and the heated lattice eventually reaches a thermal equilibrium state through thermal diffusion. This process can be described as Equation (4):(4)$$C_{l}\frac{\partial T_{l}}{\partial t}=\nabla \left(K_{l}\nabla T_{l}\right)+\frac{3NK_{b}}{\tau_{e}}\left(T_{e}-T_{l}\right)$$
where $C_{l}$ is the lattice heat capacity, $K_{l}$ is the lattice heat transfer coefficient, and $T_{l}$ is the lattice temperature [26].

The thermal radiation on the irradiated surface by the laser was considered in this numerical study, and the heat transfer of the lattice system could be described by the Fourier heat transfer, as Equation (5):(5)$$K_{l}\nabla T_{l}=\varepsilon \sigma \left(T_{sur}{}^{4}-T_{l}{}^{4}\right)$$
where ε is the thermal radiation coefficient, σ is the Stefan–Boltzmann constant and $T_{sur}$ is the ambient temperature [28]. The expression and unit of the above parameters are shown in Table 1.

## 3. Results and Discussion

### 3.1. Simulation

The simulation model of a picosecond laser heating silicon has been established, as shown in Figure 1. The laser beam in the optical path is a three-dimensional axisymmetric Gaussian beam, and the free electron concentration, electron and lattice temperature inside the material are rotationally and symmetrically distributed along the z axis. In order to improve computational efficiency and resources, the model was simplified to an r-z two-dimensional plane model. The length and width of the model were 100 μm; the initial temperature was 300 K; no constraint was imposed on the symmetry axis; the incident boundary considered the thermal radiation; and the remaining two boundaries adopted adiabatic boundary conditions, as shown in Figure 1a. Considering the accuracy and efficiency of the calculation, the grid type was set as a free quadrangle grid that was controlled by the user and calibrated as a refined semiconductor. In meshing, the geometric sequence increased along the negative direction of the r-axis and the z-axis, and the analysis domain was refined into a rectangular grid for discretization and solved at the element node, as shown in Figure 1b. In the simulation, the single pulse energy was 25 μJ; the laser wavelength was 1064 nm; the pulse widths were 15 ps, 12.5 ps, 10 ps, 7.5 ps, 6 ps and 5 ps; the analysis time step was 0.1 ps; and the total time step was 100 ps.

The solution process is shown in Figure 1c. The specific simulation steps are as follows:Step 1: Calculating the free electron concentration by Equations (1) and (2).Step 2: The electron heat capacity and electron–lattice relaxation time were calculated according to the calculated free electron concentration.Step 3: Calculating the electron temperature by Equation (3).Step 4: The electron diffusion coefficient and electron heat transfer coefficient were calculated according to the calculated electron temperature.Step 5: Calculating the lattice temperature by Equations (4) and (5).Step 6: The single photon absorption coefficient, reflection coefficient, lattice heat capacity and lattice heat transfer coefficient were calculated according to the calculated lattice temperature.Step 7: Resolving formula (2) according to the calculated reflection coefficient and single-photon absorption coefficient and repeating the above steps.

### 3.2. Simulation Results

The picosecond laser pulses have a Gaussian distribution in time, and the peak laser power density on the silicon surface is not only related to the pulse energy and spot radius, but also to the pulse width.

First, when the pulse energy was certain 25 μJ, the laser peak power density increased with the decreasing pulse width. When t_p_ = 15 ps, the laser peak power density was 2.75 × 10^15^ W/m^2^; when t_p_ = 5 ps, the laser peak power density was 8.2 × 10^15^ W/m^2^, as shown in Figure 2a. Under the action of a picosecond laser with ultrahigh power density, the electrons in the valence band of silicon absorb the photon energy and rapidly respond to the transition to the conduction band to generate free electrons [29]. A higher peak laser power density could achieve a higher free electron concentration, so the maximum free electron concentration increased with the decreasing pulse width. When t_p_ = 15 ps, the maximum free electron concentration was 2.02 × 10^27^ m^−3^; when t_p_ = 5 ps, the maximum free electron concentration was 2.76 × 10^27^ m^−3^, as shown in Figure 2b.

Second, the electron heat capacity was so small that the free electrons in the conduction band were rapidly heated in a few femtoseconds under the action of the laser-oscillating electric field. Therefore, as the peak moment of the picosecond laser power arrived, the electrons were heated to the highest temperature. A higher laser peak power density could achieve a higher electron temperature, and the maximum electron temperature increased with the decreasing pulse width. When t_p_ = 15 ps, the maximum free electron temperature was 18,713 K; when t_p_ = 5 ps, the maximum free electron temperature was 48,631 K, as shown in Figure 2c. In terms of time distribution, the free electron concentration, electron temperature and laser power density had similar variation trends. As the peak of the picosecond laser power arrived, the free electron concentration and electron temperature also rose to the maximum value and slid down to an equilibrium state after the end of the laser pulse.

Third, the time that it takes for free electrons to transfer energy to the lattice, known as electron–lattice relaxation time, is on the order of a picosecond. The electron–lattice relaxation time is a function of the free electron concentration, calculated by $\tau_{e}$ = $240\left[1+{\left(\frac{N}{6\times 10^{26}\;m^{-3}}\right)}^{2}\right]$ (fs) [19,30], and it increases with an increasing electron concentration, as shown in Figure 3a. The prolongation of the electron–lattice relaxation time leads to a longer time for the electron–lattice temperature to reach an equilibrium state. The lattice can be sufficiently heated when the electron–lattice temperature reaches an equilibrium state before the end of laser pulse, but the lattice cannot be sufficiently heated when the electron–lattice temperature reaches the equilibrium state after the end of laser pulse. The times for the electron–lattice temperature to reach an equilibrium state under the action of picosecond lasers with different pulse widths are shown in Figure 3b.

Fourth, the lattice temperature reached a maximum when t_p_ was around 7.5 ps. When t_p_ = 15 ps, the maximum free electron concentration was 2.02 × 10^27^ m^−3^, the electron–lattice relaxation time was 2.96 ps, the electron–lattice temperature reached equilibrium state at around t = 55 ps, and the laser pulse ended at t = 60 ps. Therefore, the lattice could be sufficiently heated, but the maximum electron temperature was low and the energy that was obtained by the lattice was also small. The maximum lattice temperature was 3200 K, as shown in Figure 4a. When t_p_ was shortened to 7.5 ps, the maximum free electron concentration was 2.65 × 10^27^ m^−3^, the electron–lattice relaxation time was 4.92 ps, the electron–lattice temperature reached an equilibrium state at around t = 30 ps, and the laser pulse ended at the same time (t = 30 ps). Therefore, the lattice could be sufficiently heated and the higher electron temperature enabled the lattice to reach a higher temperature. The maximum lattice temperature was 3862 K, as shown in Figure 4a. When t_p_ came to 5 ps, the maximum free electron concentration was 2.76 × 10^27^ m^−3^, the electron–lattice relaxation time was 5.31 ps, the electron–lattice temperature reached equilibrium state at around t = 22 ps, and the laser pulse ended at t = 20 ps. Although the electron temperature was the highest in this condition, the lattice could not be heated sufficiently, and the maximum lattice temperature decreased to 3749 K, as shown in Figure 4a. Therefore, when the pulse width was 5 to 15 ps, the maximum lattice temperature tended to increase first and then decrease with the decreasing pulse width, with a peak near t_p_ = 7.5 ps, as in Figure 4b.

Fifth, the optical properties of silicon, the absorption coefficient and the reflection coefficient, have a similar trend to the lattice temperature, which first increases and then decreases with the decrease in the pulse width. When t_p_ = 15 ps, the maximum absorption coefficient was 1.4 × 10^7^ m^−1^ and the maximum reflection coefficient was 0.515. When t_p_ = 7.5 ps, the maximum absorption coefficient was 2.53 × 10^7^ m^−1^ and the maximum reflection coefficient was 0.548. When t_p_ = 5 ps, the maximum absorption coefficient dropped to 2.3 × 10^7^ m^−1^ and the maximum reflection coefficient dropped slightly to 0.542, as shown in Figure 5. These were because the absorption and reflection coefficients of silicon were positively related to the lattice temperature [20,31].

Sixth, within 50 ps, vaporization and ablation of the silicon surface began to occur. In the simulation, when J = 25 μJ and t_p_ = 15 ps, the evolution of the absorption coefficient at different depths inside the silicon occurred, as shown in Figure 6a. The maximum absorption coefficient of the silicon surface was 1.4 × 10^7^ m^−1^, and the laser energy was rapidly absorbed within tens of nanometers on the silicon surface. The absorption coefficient decreased rapidly with an increasing depth. Due to the absorption of the laser energy by the silicon, the laser energy density decayed rapidly with an increasing depth. Therefore, the lattice temperature was the highest at z = 0, as shown in Figure 6b.

Figure 7 shows the two-dimensional distribution of the lattice temperature when J = 25 μJ and t_p_ = 15 ps. It can be seen that the lattice temperature presents a Gaussian distribution in space, which is due to the energy of the picosecond laser being Gaussian in space. In simulation, the vaporization ablation mechanism was adopted, and only the region where the lattice temperature was lower than the boiling point was filled in the solution domain. The picosecond laser had not yet acted on the silicon at 0 ps, and the lattice temperature was the initial temperature of 300 K. With the arrival of the laser peak, the lattice temperature increased gradually and reached the melting point of 1683.15 K at 42 ps. With the continuous heating of the picosecond laser, the boiling point of 2628.15 K was reached, and vaporization ablation began to occur on the silicon surface at 47 ps. The ablation depth of the silicon was 196 nm at 100 ps. Under the same laser parameters as the simulation, we also carried out the experiments of the picosecond laser ablation of silicon. A picosecond laser system (YPP-IR-30, HUAKUAI, Guangdong, CHINA) was selected with a pulse width of 15 ps, a maximum average power of 30 W, a maximum repetition frequency of 100 kHz and a wavelength of 1064 nm. The ablation depth was characterized using atomic force microscopy (DSM14049BF-1, Bruker, Billerica, MA, USA). The experiments were repeated five times and the ablation depths were 195.5 nm, 204.7 nm, 200.4 nm, 195.2 nm and 215.1 nm, respectively, and the average ablation depth was 202.2 nm, as shown in Table 2. There was a difference of 6.2 nm between the simulation and the experiments, and the error was less than 3.2%. The source of the error should be that the surface topography changed when the surface was ablated.

## 4. Conclusions

In this paper, a simulation model of a picosecond laser heating silicon was established based on the Fokker–Planck equation and TTM. The lattice temperature variation characteristic of silicon under the action of picosecond lasers with different pulse widths was analyzed, as well as the spatial and temporal evolution of the free electron concentration, electron temperature, reflection and absorption coefficient. When the pulse width was 5 to 15 ps, the maximum lattice temperature tended to increase first and then decrease with the decreasing pulse width, and there was a peak near t_p_ = 7.5 ps. The absorption coefficient and reflection coefficient were positively correlated with the lattice temperature, so their variation trends were similar to the lattice temperature. When processing silicon with a picosecond laser, selecting a picosecond laser with an appropriate pulse width can achieve a higher surface lattice temperature, which is beneficial for improving the processing efficiency. This numerical study has theoretical guiding significance for the selection of a picosecond laser.

## Figures and Tables

**Figure 1 micromachines-13-01119-f001:**
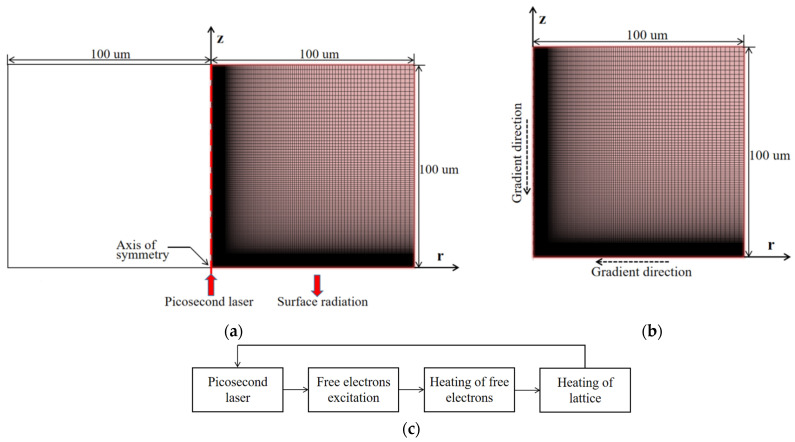
Simulation model. (**a**) boundary condition; (**b**) mesh division; (**c**) the solution process.

**Figure 2 micromachines-13-01119-f002:**
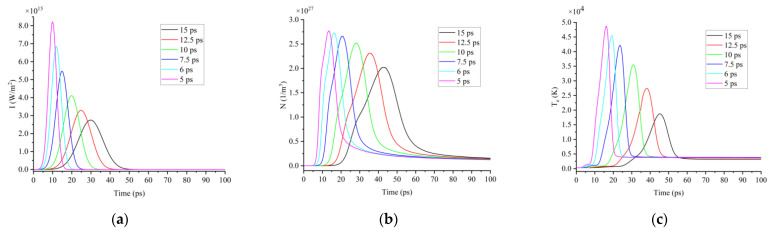
(**a**) Time distribution of laser power density; (**b**) variation characteristic of free electron concentration; (**c**) variation characteristic of electron temperature under the action of picosecond lasers with different pulse widths as 15 ps, 12.5 ps, 10 ps, 7.5 ps, 6 ps and 5 ps.

**Figure 3 micromachines-13-01119-f003:**
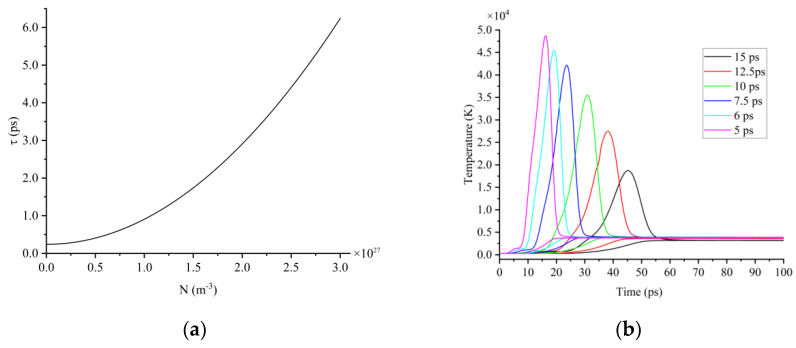
(**a**) Electron–lattice relaxation time; (**b**) time for electron–lattice temperature to reach equilibrium state under the action of picosecond lasers with different pulse widths as 15 ps, 12.5 ps, 10 ps, 7.5 ps, 6 ps and 5 ps.

**Figure 4 micromachines-13-01119-f004:**
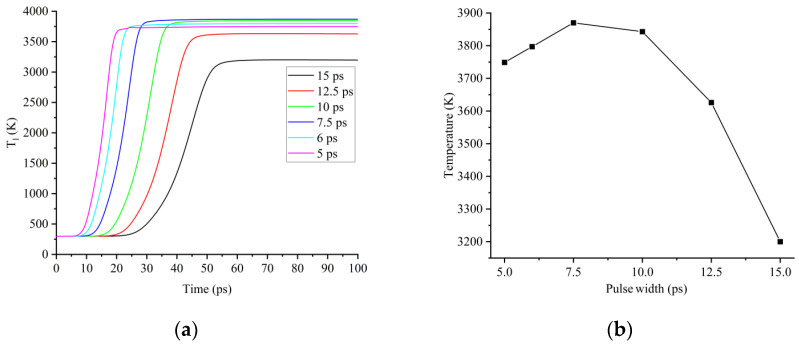
(**a**) Variation characteristic of lattice temperature; (**b**) maximum lattice temperature under the action of picosecond lasers with different pulse widths.

**Figure 5 micromachines-13-01119-f005:**
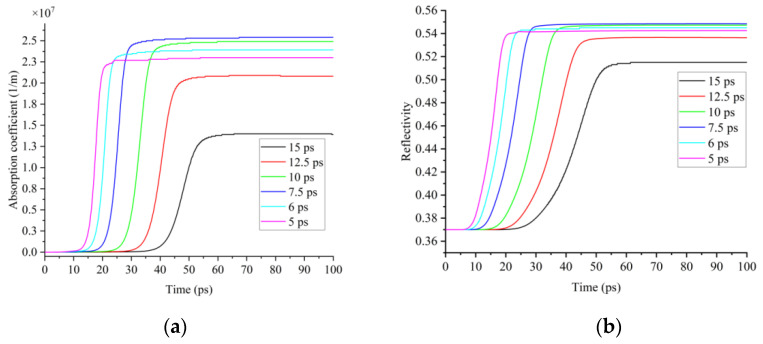
(**a**) Variation characteristic of absorption coefficients; (**b**) variation characteristic of reflection coefficients under the action of picosecond lasers with different pulse widths as 15 ps, 12.5 ps, 10 ps, 7.5 ps, 6 ps and 5 ps.

**Figure 6 micromachines-13-01119-f006:**
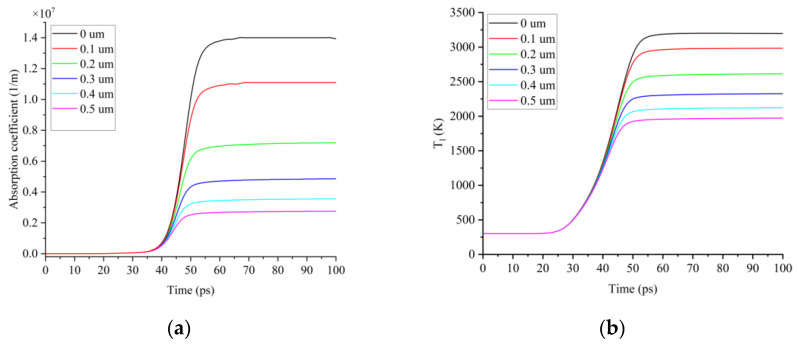
(**a**) z-axis absorption coefficient; (**b**) z-axis lattice temperature.

**Figure 7 micromachines-13-01119-f007:**
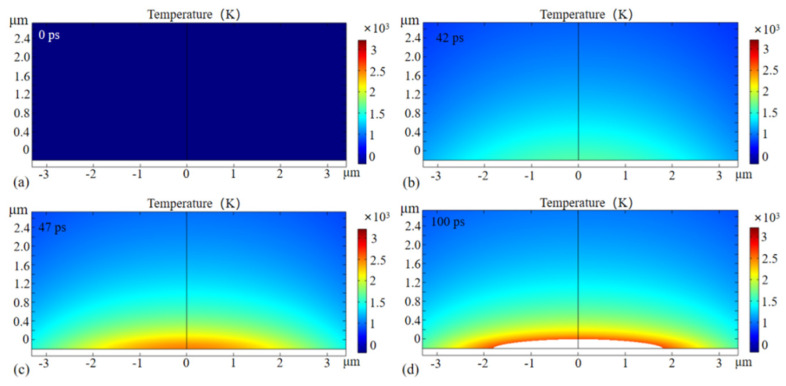
Two−dimensional lattice temperature.

**Table 1 micromachines-13-01119-t001:** Model parameters and values of silicon.

Parameter	Value	Unit	Parameter	Value	Unit
$D_{0}$ [20]	2.98 × $10^{-7}T_{e}$	m^2^/s	$\tau_{e}$ [19]	$240\left[1+{\left(\frac{N}{6\times 10^{26}\;m^{-3}}\right)}^{2}\right]$	fs
γ [27]	3.8 × 10^−43^	m^6^/s	$C_{l}$ [26]	$1.97\times 10^{6}\;+354T_{l}-3.68\times 10^{6}{T_{l}}^{-2}$	J/(m^3^·K)
α [20]	$-5895+62.26T_{l}-0.2309T_{l}$^2^ $+3.186\times 10^{-4}T_{l}{}^{3}$ + 9.967 × $10^{-8}T_{l}{}^{4}-1.409\times 10^{-11}T_{l}$^5^	1/m	$K_{l}$ [26]	$1.585\times 10^{5}T_{l}$ ^−1.23^	W/(m K)
β [20]	1.5 × 10^−11^	m/W	$r_{0}$	15	μm
R [26]	$0.37+5\times 10^{-5}(T_{l}-T_{sur}$)		$T_{sur}$	300	K
$E_{g}$ [27]	1.12	eV	ε	0.8	
$C_{e}$ [25]	$3NK_{b}$	J/(Kg·K)	σ [17]	5.67 × 10^−8^	W/m^2^·K^4^
$K_{e}$ [25]	$-0.556+7.13\times 10^{-3}T_{e}$	W/(m·K)			

**Table 2 micromachines-13-01119-t002:** The ablation depths of experiment and simulation.

No.	1	2	3	4	5	Average	Simulation
Depth	195.5 nm	204.7 nm	200.4 nm	195.2 nm	215.1 nm	202.2 nm	196 nm

## Data Availability

Data is contained within the article.

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
