# Peer review of "The Effect of Different Pulse Widths on Lattice Temperature Variation of Silicon under the Action of a Picosecond Laser"

_micromachines, 2022, doi:10.3390/mi13071119_

Round 1

Reviewer 1 Report

This manuscript reports the impact of pulse width on lattice temperature variation of silicon under the action of picosecond. Overall, the manuscript is well written. Prior to acceptance, the following minor adjustments and clarifications should be addressed:

1. Page 2, linne 52, “Maolu et al…” surname is preferred here instead of using first name. In addition, there should be a “.” after “et al”. Please double check the other cases throughout the manuscript.

2. Page 3, line 96, “spaceand”. There should be a space in between “space” and “and”.

3. Page 4, line 138, “25 μj”, should be “25 μJ”, please double check the other cases.

4. Page 5, line 161, “8.2×1015 W/m2” should be “8.2 × 1015 W/m2” (space is missing). Please double check the other cases.

5. Page 9, line 251, the authors mentioned some experiments were conducted by employing the same process parameters. It would be better if the experimental results (some figures) can be presented here, instead of only listing data. Also, it is recommended to briefly introduce the equipment and material characterization methods for the experiments.

Author Response

Response to Reviewer 1 Comments

Point 1: Page 2, line 52, “Maolu et al…” surname is preferred here instead of using first name. In addition, there should be a “.” after “et al”. Please double check the other cases throughout the manuscript.

Response 1:Thanks for your comment. I have modified it for the above question and double checked other similar questions.

Point 2: Page 3, line 96, “spaceand”. There should be a space in between “space” and “and”.

Response 2:Thanks for your comment. This is indeed a notable error, and I have corrected the error.

Point 3: Page 4, line 138, “25 μj”, should be “25 μJ”, please double check the other cases.

Response 3:Thanks for your comment. I have modified it for the above question and double checked other similar questions.

Point 4: Page 5, line 161, “8.2×1015 W/m2” should be “8.2 × 1015 W/m2” (space is missing). Please double check the other cases.

Response 4:Thanks for your comment. I have modified it for the above question and double checked other similar questions.

Point 5: Page 9, line 251, the authors mentioned some experiments were conducted by employing the same process parameters. It would be better if the experimental results (some figures) can be presented here, instead of only listing data. Also, it is recommended to briefly introduce the equipment and material characterization methods for the experiments.

Response 5:Thanks for your comment. The experimental results are not only presented in Table 2, but also presented in the description of the comparison between the experimental results and the simulation results. The equipment and characterization methods used in the experiments are listed in the last paragraph of Section 3.

Reviewer 2 Report

The manuscript with the title “The effect of different pulse width on lattice temperature variation of silicon under the action of picosecond laser” studies numerically the heating of silicon films by the interaction with picosecond pulsed laser light. The numerical results are then compared with laser ablation experiments and, apparently, there is a good match between them.

Although there is topic is of real interest, I cannot recommend the manuscript for publication. There are several reasons.

First, the paper does not bring anything new in the formal treatment with respect to literature. Second, the form of presentation is poor with many errors, besides several notorious typos like Fokker-Plangk.

In the beginning of Section 2 called “Theoretical model and parameters” the authors state - “The process of ultra-fast pulse laser heating silicon is a non-thermal equilibrium process. In semiconductor materials, the mass of electrons is much lower than that of protons”.

These statements “scratch” the sensibility of any physicist, because the pulsed laser heating is a non-equilibrium thermal process (described by Fokker-Planck equation and the other conservation equations) and always the protons are much heavier than electrons, regardless of materials.

There is an error in Eq, 2, namely the last term on the right hand side.

The table is poorly presented (the exponentials, etc.) and in addition to that the value of r0 is too small, it should be in micrometer range.  It may affect the outcome of the numerical results when compared to experiments.

In the presented set of equations the electron-lattice relaxation time is an input parameter, however the authors talk about an effective electron-lattice relaxation time. The authors should explicitly show how they calculate the effective electron-lattice relaxation time.

To summarize, the paper needs significant improvement in order to be published.

Author Response

Response to Reviewer 2 Comments

Point 1: First, the paper does not bring anything new in the formal treatment with respect to literature.

Response 1:Thanks for your comment. 

We wanted to test the conjecture that there is an optimal pulse width for a certain process target. But in the application of picosecond laser processing silicon, the pulse width of laser can not usually be changed, so it is difficult to experimentally analyze the effect of pulse width on silicon lattice temperature. The innovation of this paper is to analyze the effect of different pulse width on the lattice temperature variation of silicon through theoretical simulation. When the pulse width is 5~15ps, with the decrease of the pulse width, the maximum lattice temperature increases first and then decreases, with a peak value near tp=7.5ps.

Point 2: Second, the form of presentation is poor with many errors, besides several notorious typos like Fokker-Plangk.

Response 2:Thanks for your comment. We are so sorry!

This is indeed an obvious error, we have corrected the above errors and double-checked for other similar errors.

Point 3: In the beginning of Section 2 called “Theoretical model and parameters” the authors state - “The process of ultra-fast pulse laser heating silicon is a non-thermal equilibrium process.In semiconductor materials, the mass of electrons is much lower than that of protons”.These statements “scratch” the sensibility of any physicist, because the pulsed laser heating is a non-equilibrium thermal process (described by Fokker-Planck equation and the other conservation equations) and always the protons are much heavier than electrons, regardless of materials.

Response 3:Thanks for your comment. The statement here is really not rigorous enough, I have modified it for the above problem.

Point 4: There is an error in Eq, 2, namely the last term on the right hand side.

Response 4:Thanks for pointing out the error in equation 2. There is a typo in equation 2, the correct equation is.In the process of solving the model, the equation described the spatial and temporal distribution of laser power density. Although there is a typo in this paper, the data in this paper is obtained by the correct equation.

Point 5: The table is poorly presented (the exponentials, etc.) and in addition to that the value of r0 is too small, it should be in micrometer range. It may affect the outcome of the numerical results when compared to experiments.

Response 5:Thanks for pointing out the problem with the presentation of Table 1 and the problem of the r0 value being too small. I have modified the presentation of Table 1. There is a typo in the value of r0 in Table 1, the value of r0 is indeed in the micrometer range, and the correct value is r0 = 15μm. There is also a typo in the value of  in Table 1, the correct value is  = 3.8 × 10-43 m6/s. Although there are the above typos in this paper, the data in this paper is obtained from the above correct r0 and .

Point 6: In the presented set of equations the electron-lattice relaxation time is an input parameter, however the authors talk about an effective electron-lattice relaxation time. The authors should explicitly show how they calculate the effective electron-lattice relaxation time.

Response 6:Thanks for your comment. Electron-lattice relaxation time is a function of free electron concentration, the equation is  =(fs). This equation has been added to Table 1 and line 188 on page 7.

Round 2

Reviewer 2 Report

The manuscript has been improved. In my opinion the paper can be published.